# Quantification of Antifibrillarin (anti-U3 RNP) Antibodies: A New Insight for Patients with Systemic Sclerosis

**DOI:** 10.3390/diagnostics11061064

**Published:** 2021-06-09

**Authors:** Audrey Benyamine, Daniel Bertin, Noémie Resseguier, Xavier Heim, Julien Bermudez, David Launay, Sylvain Dubucquoi, Adrian Hij, Dominique Farge, Alain Lescoat, Isabelle Bahon-Riedinger, Nouria Benmostefa, Luc Mouthon, Jean-Robert Harlé, Gilles Kaplanski, Pascal Rossi, Nathalie Bardin, Brigitte Granel

**Affiliations:** 1Internal Medicine Department, North Hospital of Marseilles, Public Assistance Hospital of Marseilles (AP-HM), 13015 Marseilles, France; pascalmaurice.rossi@ap-hm.fr (P.R.); brigitte.granel@ap-hm.fr (B.G.); 2Aix Marseilles University (AMU), INSERM, INRA, C2VN, 13005 Marseilles, France; xavier.heim@univ-amu.fr (X.H.); Julien.bermudez@ap-hm.fr (J.B.); gilles.kaplanski@ap-hm.fr (G.K.); Nathalie.bardin@univ-amu.fr (N.B.); 3Immunology Laboratory, La Conception Hospital, Public Assistance Hospital of Marseilles (AP-HM), 13005 Marseilles, France; daniel.bertin@ap-hm.fr; 4Epidemiology and Health Economics, La Timone Hospital, AP-HM, Aix Marseilles University (AMU), 13005 Marseilles, France; noemie.resseguier@univ-amu.fr; 5Univ. Lille, U1286-INFINITE—Institute for Translational Research in Inflammation, F-59000 Lille, France; david.launay@chru-lille.fr; 6Inserm, F-59000 Lille, France; 7CHU Lille, Internal Medicine and Clinical Immunology Department, Center of Reference for Rare Autoimmune and Systemic Diseases of North and North-West France (CeRAINO), F-59000 Lille, France; 8Immunology Institute, Hospital University Center of Lille, 59037 Lille, France; Sylvain.dubucquoi@chru-lille.fr; 9Public Assistance Hospital of Paris, Saint-Louis Hospital, Autoimmune and Vascular Disease Unit, Internal Medicine (UF04), Center of reference for rare systemic autoimmune diseases (FAI2R), Université de Paris, EA 3518, Paris, France; adrian.hij@aphp.fr (A.H.); dominique.farge-bancel@aphp.fr (D.F.); 10Department of Medicine, McGill University, Montreal, QC H3G 2M1, Canada; 11Internal Medicine and Clinical Immunology Department, Hospital University Center of Rennes, 35000 Rennes, France; alain.lescoat@chu-rennes.fr; 12Immunology Laboratory, Hospital University Center of Rennes, 35033 Rennes, France; riedinger.p@wanadoo.fr; 13Internal Medicine Department, Center of reference for rare systemic autoimmune diseases of Ile de France, Cochin Hospital, Public Assistance Hospital of Paris (AP-HP), 75014 Paris, France; nouria80@yahoo.fr (N.B.); luc.mouthon@aphp.fr (L.M.); 14Internal Medicine Department, La Timone Hospital, Public Assistance Hospital of Marseilles (AP-HM), 13005 Marseilles, France; jean-robert.harle@ap-hm.fr; 15Internal Medicine and Clinical Immunology Department, La Conception Hospital, Public Assistance Hospital of Marseilles (AP-HM), 13005 Marseilles, France

**Keywords:** systemic sclerosis, autoantibodies, antifibrillarin antibodies, anti-U3 RNP antibodies

## Abstract

Background: The detection of additional autoantibodies is of great concern in systemic sclerosis (SSc) when those included in the ACR/EULAR classification are negative. In this context, the interest of antifibrillarin (anti-U3RNP) autoantibodies (AFAs) in the routine evaluation of SSc remains unclear. We aimed to assess the relevance of AFAs and their clinical association in SSc patients. Methods: In a multicenter observational retrospective study, we collected immunological and clinical data associated with AFA positivity in SSc (*n* = 42) and non-SSc patients (*n* = 13). Patients with SSc negative for AFAs (*n* = 83) were considered as a control group. AFAs were detected by indirect immunofluorescence (IIF) using HEp-2 cells, EliA or immunoblot techniques. Results: We confirmed a typical nuclear IIF pattern and showed that AFAs are mostly exclusive towards SSc conventional autoantibodies. Although also observed in non-SSc patients, high levels of AFAs with the ELiA technique allowed the diagnosis of SSc. Compared to AFA-negative SSc patients, AFA-positive SSc patients more frequently exhibited visceral involvements. They more frequently suffered from the diffuse cutaneous form and had a higher global severity of the disease. Conclusions: We demonstrate the usefulness of quantifying AFAs in the immunological exploration of SSc, especially when patients are seronegative for SSc conventional autoantibodies and display a typical IIF pattern. AFAs might constitute an interesting marker of SSc severity.

## 1. Introduction

Systemic sclerosis (SSc) is a heterogeneous disease characterized by skin and internal organ fibrosis, vascular involvement and dysregulated immunity. SSc-associated specific autoantibodies are relevant markers for the early diagnosis of the disease. The ACR/EULAR 2013 classification criteria include three antinuclear autoantibodies: antitopoisomerase 1 antibodies (ATAs), anticentromere antibodies (ACAs) and anti-RNA polymerase III antibodies (ARAs) [1]. However, about 20–25% of patients with SSc are negative for these definition markers [2,3]. In these patients, the detection of other antinuclear autoantibodies is crucial for improving the diagnosis of the disease. Among these autoantibodies, the relevance of antifibrillarin (anti-U3 RNP) antibodies in the exploration of SSc in daily practice remains to be determined.

Previous studies have described the clinical characteristics associated with AFAs in SSc patients [3,4,5,6,7,8,9,10,11,12]. The estimated frequency of AFAs ranges from 0 to 18.5% of SSc patients, with the highest prevalence found in Afro-American patients [10]. AFAs are more frequent in male patients [4,7,9] and are associated with an early onset of the disease [8,9,10], a diffuse cutaneous form [8,9,11,13] and a decreased survival [9,11]. Numerous organ involvements are related to the presence of AFAs such as pulmonary hypertension [5,6,8,9], cardiac [7,11,14], gastrointestinal tract [3,5,9,11], peripheral vascular [9] and skeletal muscle involvement [5,8,9].

Historically, the only specific method used for AFA detection was radioimmunoprecipitation, which is a cumbersome, time-consuming method [15]. More recently, the development of multiplex immunoblot [16,17,18,19] and immunoenzymatic techniques such as EliA [20,21] allows for routine AFA detection. This latter technique allows for a quantification of AFAs, but until now, the interest of such quantification in routine practice has been scarcely described in the literature.

In this retrospective study, we aimed to evaluate the usefulness of searching AFAs in the exploration of SSc in routine practice by analyzing their quantities and their association with clinical characteristics of patients.

## 2. Materials and Methods

### 2.1. Study Design

A multicenter French retrospective observational study was conducted between 2015 and 2017 in five university hospital centers (Lille, Paris St Louis, Paris Cochin, Rennes and Marseille). Fifty-five sera of patients positive for antifibrillarin antibodies (AFAs) were identified. Among these AFA-positive patients, 42 patients were suffering from systemic sclerosis (SSc) according to the ACR/EULAR 2013 classification criteria [1]. These 42 SSc patients positive for AFAs were compared for their clinical and biological characteristics to 83 SSc patients selected according to their AFA negativity in the same centers.

The scientific committee of Groupe Francophone de Recherche de la Sclérodermie Systémique (GFRS) approved the design of this study (24 November 2015). All samples were from a declared Biobank (DC 2012_1704) in compliance with ethical directives. This study has been approved by the local institutional ethics committee and has been performed in accordance with the ethical standards laid down in the 1964 Declaration of Helsinki and its later amendments. This study exclusively analyzed data from healthcare facilities, collected from medical records and centralized in one, single, fully anonymized database, which fulfilled local requirements in terms of data collection and protection of data. In accordance with French public health law (Art. L 1121-1-1, Art. L 1121-1-2), written consent from the patient is not required for this type of retrospective study. Serum samples were de-identified at the time of analysis.

### 2.2. SSc Patients

All SSc patients had a score ≥ 9 for SSc according to ACR/EULAR 2013 classification criteria [1]. Forty-two SSc patients positive for antifibrillarin antibodies (AFAs) were identified.

In each center, for each SSc patient positive for AFAs, physicians were asked to provide the clinical and biological characteristics of 2 SSc patients who were negative for AFAs. These SSc patients negative for AFAs were randomly selected among the laboratory database of each center. This allowed us then to compare 42 AFA-positive SSc patients to 83 AFA-negative SSc patients selected according to their AFA negativity in the same centers. Patients were subclassified into a limited or diffuse cutaneous form of SSc [22].

### 2.3. Non-SSc Patients Positive for Antifibrillarin Antibodies (AFAs)

Thirteen AFA-positive sera from non-SSc patients were collected from a Biobank (Marseille, DC 2012-1704), in compliance with French ethical directives in a period of four years (2012–2016).

### 2.4. Clinical Manifestations of SSc Patients

The disease duration, presence of digital ulcers, calcinosis, gastrointestinal manifestations, joint involvement and/or tendinous retraction, scleroderma renal crisis, hemoglobin levels and death were recorded. The modified Rodnan skin score (mRSS) was graded on a scale of 0–3 in 17 areas, with a maximum score of 51. Proximal muscle weakness was recorded, and the diagnosis of myositis was based on elevated CPK levels and/or the presence of signal abnormalities on muscle magnetic resonance imaging or inflammation signs on muscle biopsy sections.

Lung volumes, pulmonary capacities and single-breath carbon monoxide diffusing capacity (DLCO) corrected for hemoglobin levels results were expressed as a percentage of the predicted value. Interstitial lung disease (ILD) was diagnosed based on high-resolution computerized tomodensitometry (HRCT) imaging. Pulmonary hypertension was diagnosed by Doppler echocardiography and confirmed with right-heart catheterization. Cardiac involvement was considered based on the alteration of left-ventricular ejection fraction (LVEF) (<50%) and/or the presence of pericardial effusion.

Disease severity was measured on a scale of 0–4 according to Medsger’s severity scale [23].

### 2.5. Immunological Analyses

Antinuclear antibodies were detected by indirect immunofluorescence (IIF) on HEp-2 cells (Kallestad™ HEp-2 Cell Line Substrate, 12 wells slides, Bio-Rad Laboratories™, Hercules, CA, USA). ACAs and ATAs were detected by their typical fluorescence pattern on HEp-2 and subsequently confirmed by commercially available EliA kits (Phadia™, Uppsala, Sweden; now part of Thermo Fisher Scientific™). AFA positivity was detected by the specific EliA™ test (with a cut-off value of 10 IU/mL according to the manufacturer) or multiantigen immunoblot (EUROImmun™, Mountain Lakes, NJ, USA).

### 2.6. Statistical Analyses

Quantitative variables were described as means ± standard deviations or median and interquartile ranges (IQRs) [1st quartile–3rd quartile] according to their distribution. Qualitative variables were described as numbers and percentages.

The comparison of AFA-positive and AFA-negative SSc patients was performed using the Student’s *t*-test for quantitative characteristics and using the Chi-square test comparing proportions for nominal variables when valid (the Mann–Whitney test and Fisher’s exact test were, respectively, used otherwise, depending on the conditions of application). Receiver operating characteristic curves (ROCs) and area under the curve (AUC) with its 95% confidence interval were established. The best threshold value for AFAs to discriminate between patients with and without SSc was defined according to the Youden method. Sensitivity, specificity and calculated threshold were estimated.

Statistical analyses were performed using R software with two-sided tests. A value of *p* < 0.05 was considered significant.

## 3. Results

### 3.1. Immunological Characteristics of AFA-Positive Patients

Fifty-five patients were identified as positive for AFAs, comprising 42 SSc patients and 13 non-SSc patients. In each center, indirect immunofluorescence analysis on HEp-2 cells of AFA-positive sera showed a typical clumpy pattern consisting of a characteristic nucleolar and coilin body staining in interphase cells and reticular staining of the metaphase cells (Figure 1).

In the 13 non-SSc patients, the positivity of AFAs was confirmed by an immunoenzymatic EliA test.

In SSc patients, the positivity of AFAs was confirmed by an immunoenzymatic EliA test (*n* = 38) and immunoblot assays (*n* = 13). In the nine sera that were tested by both techniques, the median level of AFAs was 200 U/L [IQR, 178–316].

### 3.2. Demographic Characteristics of AFA-Positive Patients

The characteristics of the 42 SSc patients positive for AFAs are depicted in Table 1 and were compared to the characteristics of 83 SSc patients negative for AFAs. We observed no differences regarding the sex ratio, the age of the patients, the age at SSc diagnosis and the disease duration (Table 1).

The 13 non-SSc patients positive for AFAs were 12 women and one man with a median age of 42 years [IQR, 31–45]. They suffered from unclassified rheumatism (*n* = 4), Sjögren syndrome (*n* = 2), systemic lupus erythematosus (*n* = 1), isolated aphthous stomatitis (*n* = 1), hepatocellular carcinoma (*n* = 1), multiple sclerosis (*n* = 2) and isolated thrombophilia (*n* = 2). Among these patients, four of them were followed in our institution, and none of them developed SSc.

### 3.3. Quantification of AFA Levels in SSc and Non-SSc Patients Positive for AFA

Among AFA-positive patients, we evaluated the potential of AFA quantification to differentiate between SSc and non-SSc patients. AFA levels were available in 37 SSc patients and 13 non-SSc patients.

AFA median levels were significantly higher in SSc patients than in non-SSc patients ((Figure 2); 224 U/mL (169–316) vs. 49 U/mL (39.5–103) (*p* < 0.0001)). When AFA levels were controlled over time (between 1 and 6 years), they remained stable in both SSc patients (*n* = 7; 215 IU/mL (168–316) at baseline and 217 IU/mL (195–316) at control) and non-SSc patients (*n* = 3; 47 IU/mL (22–79) at baseline and 51 IU/mL (18–69) at control).

According to the ROC analysis, the AUC value was 0.96 (0.91–1.00), and the optimal cut-off value of AFAs associated with the diagnosis of SSc was 115 U/mL, providing a sensitivity of 91.2% and a specificity of 100.0% (Figure 3).

### 3.4. Clinical Features of AFA-Positive SSc Patients Compared with AFA-Negative SSc Patients

To investigate whether the presence of AFAs was associated with a particular profile of the disease, AFA-positive patients with SSc (*n* = 42) were compared to AFA-negative patients with SSc (*n* = 83) (Table 1).

Only two AFA-positive patients exhibited concomitant positivity for classical SSc autoantibodies, namely ACAs (*n* = 2). These two patients exhibited low levels of AFAs under the cut-off value of 115 U/mL. In AFA-negative patients, ATAs and ACAs were the two main antibodies. Other autoantibodies such as anti-DNA, anti-SSA, anti-SSB, anti-PMScl and anti-U1 RNP were less frequent in AFA-positive SSc patients than in AFA-negative patients (*p* = 0.048).

In the AFA-positive group, a higher proportion of the patients had diffuse cutaneous SSc (*p* = 0.02) with a higher mean mRSS (*p* = 0.02). Muscle involvement was also more frequent in AFA-positive patients (*p* = 0.004).

AFA-positive patients more often had lower digestive tract involvement with chronic intestinal pseudo-obstruction (*p* = 0.0003) and diarrhea (*p* = 0.003).

AFA-positive patients more frequently complained of dyspnea (62.5% vs. 39.8%; *p* = 0.02). They exhibited a lower mean forced vital capacity (*p* = 0.02) and a DLCO <70% was more frequently observed in AFA-positive patients (*p* = 0.01).

Concerning cardiac involvement, AFA-positive patients had a lower mean LVEF (*p* = 0.0256) and a higher frequency of pericardial effusion (*p* = 0.0273). AFA-positive patients with myositis have more frequently LVEF below 50% compared with AFA-positive patients without myositis (*p* = 0.01).

### 3.5. Clinical Features of AFA-Positive SSc Patients According to the Cutaneous Form of the Disease

As AFA-positive SSc patients more frequently exhibited a diffuse cutaneous form, we wondered whether severe organ involvement was associated with diffuse cutaneous SSc or with AFA positivity. Hence, in order to further assess the clinical characteristics of AFA-positive patients, we performed a matched comparison according to the cutaneous form.

The 24 AFA-positive SSc patients with a diffuse cutaneous form were compared to the 28 AFA-negative SSc patients with a diffuse cutaneous form. We showed that patients with AFAs more often suffered from myositis (34.78% vs. 11.54%; *p* = 0.0516), diarrhea (47.62% vs. 18.52%; *p* = 0.0309) and chronic intestinal pseudo-obstruction (33.3% vs. 3.7%; *p* = 0.0147). DLCO was more often under the cut-off value of 70% of the predicted value in AFA-positive SSc patients (90.91% vs. 59.26%; *p* = 0.0126).

The 17 AFA-positive SSc patients with a limited cutaneous form were compared with the 52 AFA-negative SSc patients with a limited cutaneous form. We observed that patients with AFAs were more often male (35.29% vs. 9.62%; *p* = 0.0208). AFA-positive patients more often exhibited digital ulcers (50% vs. 21.57%; *p* = 0.0531) and chronic intestinal pseudo-obstruction (17.65% vs. 4%; *p* = 0.0992). We found that AFA-positive patients more frequently complained of dyspnea (81.25% vs. 38.46%; *p* = 0.0027), had a lower mean FVC (85.8 vs. 99.8; *p* = 0.033) and interstitial lung disease on HRCT (60% vs. 20%; *p* = 0.0397).

### 3.6. Disease Severity of AFA-Positive Patients Compared with AFA-Negative SSc Patients

AFA-positive patients exhibited higher severity of the disease as shown by a higher mean Medsger score (2.9 vs. 2.3; *p* = 0.0004) and a higher proportion of patients with severe involvement attested by a Medsger score of 4 (25% vs. 8.5%; *p* = 0.008).

When patients were matched according to their cutaneous form, AFA-positive patients still exhibited a higher severity than AFA-negative patients in an independent manner from the limited cutaneous form (mean Medsger score of 2.82 vs. 2.27; *p* = 0.0461) or diffuse cutaneous form (2.95 vs. 2.41; *p* = 0.0491). During the study period, a higher proportion of deceased patients was found in the AFA-positive group (16.67% vs. 2.38%; *p* = 0.0547).

## 4. Discussion

In the present study, the reading of the IIF pattern of positive AFAs on HEp-2 cells confirmed the typical pattern of clumpy nucleolar and coilin body staining [8,24]. Deciphering this specific pattern by skilled readers is crucial to trigger subsequent biological exploration and confirm AFA positivity. Indeed, a previous study has shown an excellent agreement between this peculiar IIF pattern and the positivity of AFAs using the historical gold standard technique, namely radioimmunoprecipitation [8].

We demonstrated here the value of using a quantitative method to confirm IIF screening because high AFA levels were 100% specific for SSc. As previously described, we found AFAs in patients with other autoimmune diseases [8] and solid tumors [16,25]. In all these cases, we showed that AFA levels were under the specific cut-off value of 115 U/mL determined for SSc according to the EliA test from Phadia™.

In the context of SSc, AFA positivity usually appears exclusive towards the classical SSc autoantibodies (ATAs, ACAs, ARAs). The negativity of SSc-associated specific autoantibodies makes the diagnosis more challenging and can delay specific medical management. Hence, we recommend searching for AFAs when IIF screening displays the typical clumpy nucleolar immunofluorescence pattern of AFAs, in patients with clinical suspicion of SSc and a negative search for classical SSc autoantibodies. In these patients, the confirmation of AFAs should be performed either by immunoblot [17] or by immunoenzymatic technique when available. In our study, we observed a good agreement between those two techniques. While the first allows for the detection of other nucleolar patterned antinuclear antibodies such as anti-PM Scls or anti-Th/To autoantibodies, the second enables the quantification of AFAs but requires specific equipment.

Associations between AFAs and ethnicity via HLAs have been suggested [26], supporting the crucial need to characterize these antibodies on homogeneous groups of patients regarding geographical origin. Most of the studies have been conducted in America. A few European studies [8,13,27] and one French study, including 37 AFA SSc-positive patients, have been reported [28]. This latter study identified a younger age at SSc disease onset and a predominance of male patients, as well as a higher Rodnan skin score and a higher frequency of myositis in the AFA-positive population. In line with these findings, we observed an association between AFAs and the diffuse cutaneous form of the disease and muscle involvement. In our study, we evidenced more organ involvement, notably the digestive tract with a manifestation of chronic intestinal pseudo-obstruction, interstitial lung disease and heart involvement, consisting of systolic dysfunction and pericardial effusion in AFA-positive patients. Interestingly, heart involvement was all the more frequent in AFA-positive patients with myositis.

Our study underlines that AFAs are associated with a severe form of SSc disease assessed by a higher global Medsger severity score. This association of AFAs with disease severity was independent of the cutaneous form of the disease and remained significant when patients were matched according to their cutaneous form. We also observed a higher proportion of deceased patients in AFA-positive SSc patients, which is consistent with the decreased survival in AFA-positive patients reported in the literature [9,11].

In conclusion, our data demonstrate the relevance of detecting and quantifying AFAs in the immunological exploration of SSc, especially when conventional autoantibodies are negative and with a typical IIF antinuclear clumpy nucleolar pattern. Our study brings up a new insight into the diagnosis of SSc by introducing the relevance of the AFA quantification. We also underscore that AFAs appear as an interesting marker of the severity of the disease. Prospective longitudinal studies are now required to evaluate the potential use of AFAs as a prognostic marker.

## Figures and Tables

**Figure 1 diagnostics-11-01064-f001:**
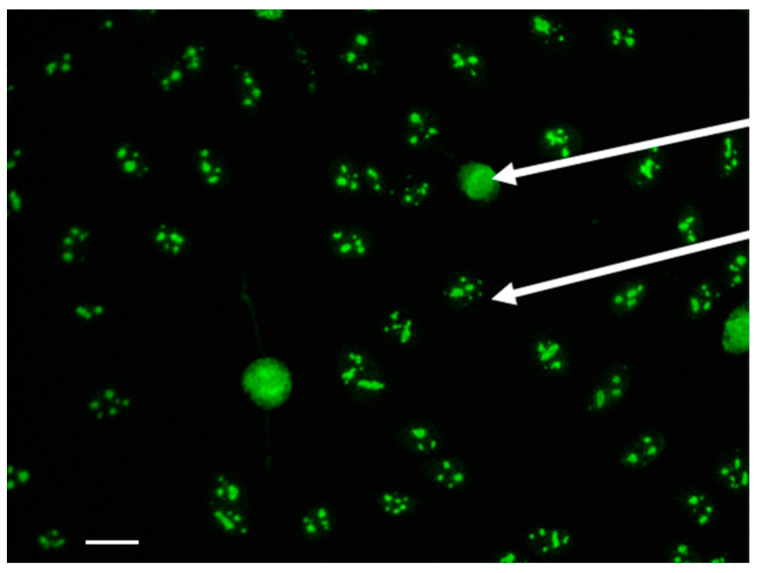
Typical fluorescence pattern of antifibrillarin antibodies by indirect immunofluorescence on HEp-2 cells. The upper arrow depicts the reticular staining of a metaphase cell. The lower arrow shows clumpy nucleolar staining with fluorescent coilin bodies in an interphase cell. Scale bar: 20 µm

**Figure 2 diagnostics-11-01064-f002:**
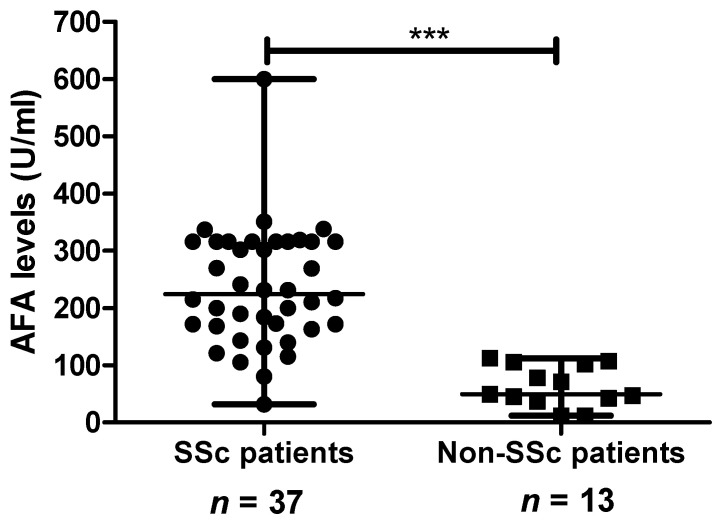
Comparison of antifibrillarin antibody (AFA) levels in AFA-positive systemic sclerosis (SSc) patients (*n* = 37) and AFA-positive non-SSc patients (*n* = 13). AFA levels were assessed with EliA. Results are expressed as median ± interquartile range. *** *p* < 0.0001.

**Figure 3 diagnostics-11-01064-f003:**
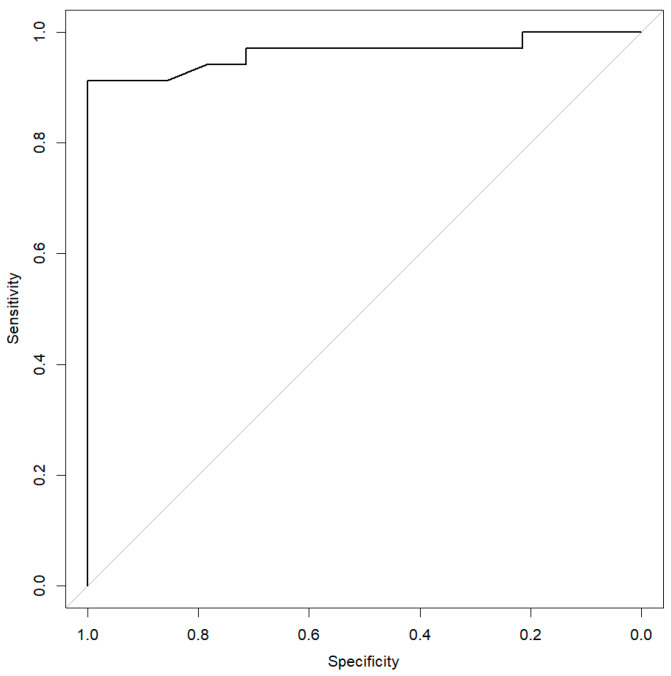
Receiver operating characteristic curve (ROC) analysis. Sensitivity and specificity of antifibrillarin antibody (AFA) levels for the diagnosis of SSc were optimal for a cut-off value of 115 U/mL.

**Table 1 diagnostics-11-01064-t001:** Comparison of clinical and immunological characteristics between antifibrillarin autoantibody (AFA)-positive and AFA-negative SSc patients.

SSc Patients’ Characteristics	AFA-Positive SSc Patients (*n* = 42)	AFA-Negative SSc Patients (*n* = 83)	*p*-Value
**Demographics:**			
Female/Male	9/33	15/68	0.6527 ^a^
Age (years, mean ± SD)	51.3 ± 13.4	56.14 ± 15.7	0.0746 ^c^
Age at SSc diagnosis (years, mean ± SD)	46.7 ± 14.9	49.3 ± 14.8	0.3656 ^c^
Disease duration (months, mean ± SD)	67.5 ± 57.3	83.6 ± 85.8	0.2734 ^c^
**SSc classification and scores:**			
Diffuse/limited/sine scleroderma SSc (*n, %)*	24 (58.5)/17 (41.5)/0	28 (34.1)/52 (63.4)/2 (2.4)	**0.0220 ^b^**
mRSS (mean ± SD)	14.9 ± 10.3	9.8 ± 9.5	**0.0126 ^c^**
Medsger’s severity scale score/4 (mean ± SD)	2.9 ± 0.8	2.3 ± 1	**0.0004 ^c^**
**Cutaneous/musculoskeletal involvement:**			
Digital ulcers (*n*, %)	17 (43.6%)	22 (26.8%)	0.0652 ^a^
Calcinosis (*n*, %)	10 (28.6%)	17 (23.3%)	0.5528 ^a^
Joint involvement (*n*, %)	20 (47.6%)	41 (50.6%)	0.7525 ^a^
Myositis * (*n*, %)	10 (25,0%)	5 (6.4%)	**0.0041 ^a^**
**Gastrointestinal involvement:**			
Gastroesophageal reflux (*n*, %)	29 (72.5%)	52 (62.7 %)	0.2805 ^a^
Chronic intestinal pseudo-obstruction (*n*, %)	11 (28.2%)	3 (3.8%)	**0.0003 ^b^**
Diarrhea (*n*, %)	16 (41.0%)	13 (16.5%)	**0.0035 ^a^**
**Pulmonary involvement:**			
Dyspnea (*n*, %)	25 (62.5%)	33 (39.8%)	0.0179 ^a^
FVC < 70% of predicted (*n*, %)	9 (22.5%)	7 (9.1%)	**0.0452 ^a^**
TLC < 80% of predicted (*n*, %)	11 (26.8%)	15 (20.0%)	0.3991 ^a^
DLCO < 70% of predicted (*n*, %)	33 (84.6%)	48 (62.3%)	**0.0135 ^a^**
Interstitial lung disease on HRCT	23 (47.4 %)	18 (29.1%)	0.0526 ^a^
PAH (right-heart catheterization) (*n*, %)	12 (29.3%)	12 (16.7%)	0.1153 ^a^
**Cardiac involvement:**			
LVEF (%, mean ± SD)	61.7 ± 10.1	65.7 ± 7.6	**0.0256 ^c^**
Pericardial effusion (*n*,%)	9 (25.7%)	7 (9.6%)	**0.0273 ^a^**
Abnormal EKG (*n*, %)	9 (24.3%)	19 (30.7)	0.4993 ^a^
MRI Late myocardial gadolinium enhancement (*n*, %)	4 (33.3%)	4 (28.6%)	1.0000 ^c^
**Renal involvement:**			
Scleroderma renal crisis (*n*, %)	1 (2.4%)	0 (0%)	0.3471 ^b^
**SSc-associated autoantibodies:**			
Antitopoisomerase 1 (*n, %*)	0	27 (35.1%)	**0.0001 ^a^**
Anticentromere (*n, %*)	2 (5.4%)	29 (36.7%)	**0.0004 ^a^**
Anti-RNA polymerase III antibodies (*n, %*)	0	2 (3.1%)	1.0000 ^b^
**Other autoantibodies (*n*, %):**	13 (17%)	13 (33%)	**0.0489** ^a^
-anti-DNA (*n, %*)	0	1 (1.3%)	-
-anti-Ro/SSA (*n, %*)	7 (10.3%)	7 (9.2%)	-
-anti-La/SSB (*n, %*)	3 (7.7%)	0	-
-anti-PM-Scl (*n, %*)	2(5.4%)	1(1.3%)	-
-anti-U1 RNP (*n, %*)	1 (2.6%)	3 (4.0%)	-
**Other biological results:**			
Hemoglobin (g/dl)	12.2 (±1.7)	13.1 ((±1.6)	**0.0042 ^c^**
Antinuclear antibodies positivity (*n,* %)	42 (100%)	82 (100%)	N/A

mRSS: modified Rodnan skin score; DLCO: single-breath diffusing capacity of the lung for carbon monoxide (CO); KCO: carbon monoxide transfer coefficient; HRCT: high-resolution computed tomography; EKG: electrocardiogram; MRI: magnetic resonance imaging; AFA: antifibrillarin autoantibodies; FVC: forced vital capacity; TLC: total lung capacity; DLCO: diffusing capacity of the lung for carbon monoxide; PAH: pulmonary arterial hypertension; LVEF: left-ventricular heart fraction; CPK: creatine phosphokinase; * The diagnosis of myositis was based on elevated CPK levels and/or the presence of signal abnormalities on muscle MRI or inflammation signs on muscle biopsy; ^a^: *p*-value from the Chi-square test; ^b^: *p*-value from Fisher’s exact test; ^c^: *p*-value from the Student’s *t*-test; *p*-values <0.05 appear in bold format.

## Data Availability

The data presented in this study are available on request from the corresponding author.

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
