# Peer review of "Quantification of Antifibrillarin (anti-U3 RNP) Antibodies: A New Insight for Patients with Systemic Sclerosis"

_diagnostics, 2021, doi:10.3390/diagnostics11061064_

Round 1

Reviewer 1 Report

This  is  a  nice  study  looking  at  anti Fibrillarin  antibodies in  patients with systemic  sclerosis.   I  have  several comments and  suggestions  that  would have  strengthen  the  paper.

  1. Introduction: although most  patients with  scleroderma  have  one of the  3  primary  scleroderma  specific  antibodies,   there are many  with  the  lesser  common   scleroderma  antibodies  including  U3 RNP or  anti fibrillarin,  I  would NOT call them  seronegative SSc  subgroup.   There  are  a much  smaller number  who actually  are  seronegative  without  even an ANA  and  those  patients  should  be designated as seronegative,   These  patients  are  nucleolar  patterned +ANA. 
  2. Intro: Actually  I  think  the  classic  gold  standard of  defining  this antibody is  by  immunoprecipitation,   and  the  newer  multiplex immunoblot and immunoenzymatic  techniques  have  made  it  easier  to  get  these  results since  the immunoprecipitation   is  not  easily  obtained  clinically.   However,  I do not  think  there have  been studies  that have  confirmed  these  methods  in comparison  to  the  gold standard,   My  limited  experience  as  we  have  just  had  one  immunoblot  test  available  over  the past  couple  of years,  is  that they  are  good  but  are  still not  identifying  all the  nucleolar patterned  ANAs in scleroderma. 
  3. Study design:  Unfortunately  I  am  most  bothered  by  your  comparison population,  One  you don’t  give  us  any  idea  of how many  patients (or samples)  were  obtained to identify  these  53 patients, or  even  the  42 SSc patients, so  we  don’t have an idea  of  frequency.   However,  more  bothersome,  is  the way  the  additional  82 patients  were  identified  or  actually  weren’t  identified.   If you  wanted  to  identify  the organ  involvements  in these  patients  compared  to other scleroderma  patients,  it  might  have  been  better  to match  them  at  least in general  by  sex,  age, disease  duration and disease  subset .  At  least  that  way,  although you  couldn’t  tell that they may have  been more male and more diffuse,   you might have  been able to better  clarify  the organ  involvement.   Because your  comparison group  has  significantly  fever  diffuse  cutaneous  disease,  it  is  harder  to interpret  your  clinical  findings.  Do they have more severe  disease just because they are  more  diffuse?  I  concur  that  your  findings  are  very  typical  for this subset of patients  but  you  could have  had much stronger and meaningful   evidence  by  doing a  little match.   What  would have  been  best  is if you had  80  patients  with a nucleolar  pattern ANA  that were NOT  fibrillarin  although  that  would  not  likely be  possible. 
  4. You don’t mention  race,  were  these all  Caucasian  patients?
  5. You actually   had the  3  patients  who had topo and  ACA  as  well as  2  more  with  PmScl  and  1  with U1 RNP  or a  total of 6 or  42%  who have  more  the one  scleroderma  specific  antibody,   Because of these immunoblot assays  pick up  low  levels,  just  because  they  have a  very slightly  positive  test  doesn’t  mean they are  clinically  relevant.      A  patient  with a high  titer of one  and a  very low  titer of the  other is  more  likely  to have  clinical associations  with the high  titer  one,   Does that change  anything?  Particularly  since you  have  a  few  SSc  with low  titers  of  the AFA , Are  they  the  ones  with high titers  of other antibodies???
  6. Please emphasize  the clumpy  nucleolar  pattern  and encourage  reader  if they  have a  nucleolar  pattern with neg ACA,  topo or  RNA pol III,  they  really should  get  an immunoblot, or  if  this  immunoenzyme is available.  The  immunoblot  will also  pick up the other nucleolar  pattern ANAs,  in cluding several PmScls  and the  Th/To  antibody,   
  7. I don’t  think  everyone  has  access  to  this immunoenzymatic  assay  from Phadia   so you  might  acknowledge  that, 

Reviewer 2 Report

I rad with interest the paper by Benyamine and colleagues about the anti-U3-RNP antibodies in SSc patients. I have some major methodological issues about this paper:

1) I think that It would be appropriate to double check the positivity for anti-U3-RNP as different methods were used for the detection and a few patients have a double positivity for other SSc Abs (1 ATA, 2 ACA 2 Pm-Scl, 1 U1-RNP) as this is unusual and the paper is focused on the positivity of anti-U3-RNP Ab this is the strongetst weakness of the paper

2) In Table 1 77 patients were positive for ANA but the AFA negative choort included 83 patients. Were 6 patients not tested for ANA? 

3) I would be interested in the how many patients diagnosed with myositis had imaging or bioptic or electromyographic confirmation beside the presence of elevated CPK

4) Did all patients undergo RCH?

Author Response

Response to Rewiever n°2

I rad with interest the paper by Benyamine and colleagues about the anti-U3-RNP antibodies in SSc patients. I have some major methodological issues about this paper:

1) I think that It would be appropriate to double check the positivity for anti-U3-RNP as different methods were used for the detection and a few patients have a double positivity for other SSc Abs (1 ATA, 2 ACA 2 Pm-Scl, 1 U1-RNP) as this is unusual and the paper is focused on the positivity of anti-U3-RNP Ab this is the strongetst weakness of the paper

The positivity for anti-U3 RNP antibodies was confirmed by at least 2 techniques in 39/42 patients, consisting in a first screening by IIF revealing the typical AFA clumpy nucleolar pattern and a confirmation either by an immunoblot technique (n= 4) or an EliA test (n=26).

The positivity for anti-U3 RNP antibodies was confirmed by the 3 techniques namely IIF analysis, immunoblot technique and EliA technique in 9 patients. This point was added in the results section of the revised version of the manuscript in lines 173-175.

Only 3 patients had a positive search for AFA by only one technique namely EliA. In these patients, typical pattern of AFA was not found as IIF patterns were: speckled n=2, speckled and centromeric n=1. These patterns were found in AFA positive patients with co-positivity of anti-SSA (n=1), anti-SSA/anti-SSB (n=1) and anti-centromere/anti-SSA (n=1).

2) In Table 1 77 patients were positive for ANA but the AFA negative choort included 83 patients. Were 6 patients not tested for ANA? 

The AFA negative cohort included 83 patients with a positive search of ANA by Indirect Immunofluorescence analysis in 82 patients. We modified the Table 1 accordingly. We have missing data regarding the positivity of ANA regarding 1 patient. However, in this patient, the search for AFA was negative with immunoblot technique (n=1), that is the reason why this SSc patient was included in the study despite this missing data. 

3) I would be interested in the how many patients diagnosed with myositis had imaging or bioptic or electromyographic confirmation beside the presence of elevated CPK

Fifteen SSc patients were diagnosed with myositis. Among these patients, 8 patients had a muscle histological confirmation of myositis, 3 patients had an imaging confirmation by magnetic resonance imaging. Five patients have elevated CPK associated with muscular weakness. One patient had an electromyographic confirmation.

4) Did all patients undergo RCH?

Not all patient underwent a Right Heart Catheterization. Right Heart Catheterization was performed when pulmonary arterial hypertension was suspected based on the results of Doppler echocardiography. Twenty-four patients had a pulmonary arterial hypertension confirmed by Right Heart Catheterization.

Reviewer 3 Report

This manuscript by Benyamine et al describes the quantification of serum anti-fibrillarin autoantibodies (AFA) and highlights relationships between AFA and disease characteristics as well as between AFA and other anti-nuclear autoantibodies in patients with systemic sclerosis (SSc). The overall conclusions are that (1) a commercially available enzyme-linked immunoassay (EliA) for AFA has high sensitivity and specificity for SSc, (2) AFA positivity is associated with relatively severe forms of SSc with multi-organ involvement, and, (3) AFA positivity is more frequently found in SSc patients negative for other “disease marker” anti-nuclear antibodies. This reviewer feels that the authors need to address some issues and make corrections as suggested below:

  1. While the results of this study may be of interest from a clinical standpoint, the novelty of the findings is limited. For example, it is unclear if this paper provides any new information that add to the results of Tall et al (Ref. #24), using similar approaches and reporting relationships between AFA positivity and clinical characteristics in patients with SSc. If there are novel results in the present manuscript, as compared to the study published by Tall et al, they must be emphasized in the Discussion.
  2. There is an issue with the statistical analysis. The authors used different statistical tests for the analysis of “quantitative variables” and “qualitative characteristics”. Although the analytical approaches are correct, they rendered numeric values (n and %) to the “qualitative” data. It would be more appropriate to use the term “nominal variables” for the “qualitative characteristics”. The authors also need to clarify that they used the percentages (%) to calculate the p values and identify (by a, b c, or by symbols) each of the specific statistical test employed to compare the variables of the AFA positive and AFA negative groups in Table 1.
  3. Line 36: “routine exploration of SSc” should be replaced by “routine evaluation of SSc”.
  4. Line 63: “remains to determine” should be replaced by “remains to be determined”.
  5. Line 77: “analyzing their quantification” should be replaced “analyzing their quantities”.
  6. Line 95: “from healthcare” should be followed by “facilities”.
  7. Line 99: After the last sentence, it should be stated if the serum samples were de-identified at the time of analysis.
  8. Line 121: “muscle weakness was notified” should be replaced by “muscle weakness was recorded”.
  9. Line 123: “signs on muscle biopsy” should be followed by “sections”.
  10. Line 174: “a” should be removed from “a SSc”.
  11. Line 178: “we evaluated the interest of the AFA quantification” should be replaced by “we evaluated the potential of AFA quantification”.
  12. Lines 206 and 210: The abbreviations DLCO and LVEF should be spelled out in the text (not only in the footnote to Table 1).
  13. Line 218: The proportion (at least %) of deceased patients should be provided for both the AFA-negative and AFA positive groups.
  14. Line 244: The meaning of “the centralized reading” should be clarified.
  15. Lines 247 and 248: “the interest to use” should be replaced by “the value of using”.
  16. Line 263: “were associated in AFA positive patients” should be replaced by “were associated with AFA positivity in SSc patients”.
  17. Line 271: “a potential interest of AFA” should be replaced by “the potential use of AFA”.

Author Response

Response to Rewiever n°3

This manuscript by Benyamine et al describes the quantification of serum anti-fibrillarin autoantibodies (AFA) and highlights relationships between AFA and disease characteristics as well as between AFA and other anti-nuclear autoantibodies in patients with systemic sclerosis (SSc). The overall conclusions are that (1) a commercially available enzyme-linked immunoassay (EliA) for AFA has high sensitivity and specificity for SSc, (2) AFA positivity is associated with relatively severe forms of SSc with multi-organ involvement, and, (3) AFA positivity is more frequently found in SSc patients negative for other “disease marker” anti-nuclear antibodies. This reviewer feels that the authors need to address some issues and make corrections as suggested below:

  1. While the results of this study may be of interest from a clinical standpoint, the novelty of the findings is limited. For example, it is unclear if this paper provides any new information that add to the results of Tall et al (Ref. #24), using similar approaches and reporting relationships between AFA positivity and clinical characteristics in patients with SSc. If there are novel results in the present manuscript, as compared to the study published by Tall et al, they must be emphasized in the Discussion.

We thank the reviewer for his comment. Our paper provides new information in addition to the study of Tall et al. regarding the clinical characteristics of AFA patients and the quantification of AFA as a useful tool for the diagnosis of SSc.

Tall et al. studied the clinical features of 37 AFA positive SSc patients identified by IIF screening and by an in-house Western Blot technique and/or the PHADIA commercially available EliA test. They identified a younger age at SSc disease onset, a predominance of male patients in the AFA positive population as well as a higher Rodnan skin score and a higher frequency of myositis.

In our study, we found no difference regarding sex ratio and age at SSc disease onset. Regarding organ involvement, we found an association between AFA and the diffuse cutaneous form of the disease and confirmed the association with a higher Rodnan score and with muscle involvement. In our study but not in that of Tall et al., AFA-positive patients had more often lower digestive tract involvement and pulmonary involvement.  We also described that AFA-positive patients exhibited higher severity of the disease as shown by a higher mean Medsger score (2.9 versus 2.3; p=0.0004) and a higher proportion of patients with severe involvement attested by a Medsger score of 4 (25% versus 8.5%; p=0.008).

In our study, one of the innovative aspects is the quantitative analysis of AFA levels. We found that patients with SSc had higher levels of AFA compared with patients who had no diagnosis of SSc. We identify a cut-off value of 115 IU/l which is associated with the SSc diagnosis with a high sensitivity and a high specificity.

As requested by the reviewer, we added a part in the discussion (lines 307-325) and in the conclusion (lines 337-340) in order to emphasize the innovative aspects of our study.

  1. There is an issue with the statistical analysis. The authors used different statistical tests for the analysis of “quantitative variables” and “qualitative characteristics”. Although the analytical approaches are correct, they rendered numeric values (n and %) to the “qualitative” data. It would be more appropriate to use the term “nominal variables” for the “qualitative characteristics”. The authors also need to clarify that they used the percentages (%) to calculate the p values and identify (by a, b c, or by symbols) each of the specific statistical test employed to compare the variables of the AFA positive and AFA negative groups in Table 1.

We apologize for this mistake and replaced the term “qualitative characteristics” by “nominal variables” in the revised version of the manuscript in line 154.

We clarified that we used proportions to calculate p values in line 154.

As requested by the reviewer, we identified (by a, b c) each of the specific statistical test employed to compare the variables of the AFA positive and AFA negative groups in the revised version of Table 1 and in the legend (lines 267-268) where “a”, “b” and “c” respectively stand for “chi-square test”, “Fisher test” and “Student t-test”.

  1. Line 36: “routine exploration of SSc” should be replaced by “routine evaluation of SSc”.

We modified the text accordingly in line 36.

  1. Line 63: “remains to determine” should be replaced by “remains to be determined”.

We modified the text accordingly in line 63.

  1. Line 77: “analyzing their quantification” should be replaced “analyzing their quantities”.

We modified the text accordingly in line 80.

  1. Line 95: “from healthcare” should be followed by “facilities”.

We modified the text accordingly in line 98.

  1. Line 99: After the last sentence, it should be stated if the serum samples were de-identified at the time of analysis.

Serum sample were de-identified at the time of analysis. We add this statement accordingly in lines 102-103.

  1. Line 121: “muscle weakness was notified” should be replaced by “muscle weakness was recorded”.

We modified the text accordingly in line 127.

  1. Line 123: “signs on muscle biopsy” should be followed by “sections”.

We add the term “section” accordingly in line 129.

  1. Line 174: “a” should be removed from “a SSc”.

The correction was made accordingly in line 186.

  1. Line 178: “we evaluated the interest of the AFA quantification” should be replaced by “we evaluated the potential of AFA quantification”.

The correction was made accordingly in line 190.

  1. Lines 206 and 210: The abbreviations DLCO and LVEF should be spelled out in the text (not only in the footnote to Table 1).

The abbreviation DLCO was spelled out in lines 130-131, in the material and methods section. According to the reviewer comment, we spelled out the abbreviation LVEF in the material and methods section in line 136.

  1. Line 218: The proportion (at least %) of deceased patients should be provided for both the AFA-negative and AFA positive groups.

We provide the percentage of deceased patients for each group in the revised version of the manuscript in lines 257-258.

  1. Line 244: The meaning of “the centralized reading” should be clarified.

We apologize for the misused term “centralized reading”. Each clinical reference centre had a medical biologist referee for ANA screening its corresponding reference laboratory. We replaced this sentence by « In each center, the indirect immunofluorescence analysis on HEp-2 cells of AFA positive sera showed …” in line 167.

  1. Lines 247 and 248: “the interest to use” should be replaced by “the value of using”.

We modified the text accordingly in line 290.

  1. Line 263: “were associated in AFA positive patients” should be replaced by “were associated with AFA positivity in SSc patients”.

“Interestingly, heart and peripheral muscle involvement were associated in AFA-positive patients”. In this sentence which may have been misleading, we meant to specify that cardiac involvement was more frequent in AFA positive patients with myositis. We modified this sentence accordingly in lines 318-319

  1. Line 271: “a potential interest of AFA” should be replaced by “the potential use of AFA”.

We modified the text accordingly in lines 340-341.

Round 2

Reviewer 2 Report

No further comments

Reviewer 3 Report

The authors addressed all of the Reviewers' concerns and revised the manuscript accordingly.